# A Rapid and Simple UHPLC-MS/MS Method for Quantification of Plasma Globotriaosylsphingosine (lyso-Gb3)

**DOI:** 10.3390/molecules26237358

**Published:** 2021-12-03

**Authors:** Alessandro Perrone, Susan Mohamed, Vincenzo Donadio, Rocco Liguori, Manuela Contin

**Affiliations:** 1IRCCS Istituto delle Scienze Neurologiche di Bologna, 40139 Bologna, Italy; alessandro.perrone@ausl.bologna.it (A.P.); susan.mohamed2@unibo.it (S.M.); vincenzo.donadio@unibo.it (V.D.); rocco.liguori@unibo.it (R.L.); 2Department of Biomedical and Neuromotor Sciences, University of Bologna, 40139 Bologna, Italy

**Keywords:** lyso-Gb3, UHPLC-MS/MS, Fabry disease, protein precipitation

## Abstract

Fabry disease (FD) is a rare X-linked lysosomal storage disorder caused by α-galactosidase A gene (GLA) mutations, resulting in loss of activity of the lysosomal hydrolase, α-galactosidase A (α-Gal A). As a result, the main glycosphingolipid substrates, globotriaosylceramide (Gb3) and globotriaosylsphingosine (lyso-Gb3), accumulate in plasma, urine, and tissues. Here, we propose a simple, fast, and sensitive method for plasma quantification of lyso-Gb3, the most promising secondary screening target for FD. Assisted protein precipitation with methanol using Phree cartridges was performed as sample pre-treatment and plasma concentrations were measured using UHPLC-MS/MS operating in MRM positive electrospray ionization. Method validation provided excellent results for the whole calibration range (0.25–100 ng/mL). Intra-assay and inter-assay accuracy and precision (CV%) were calculated as <10%. The method was successfully applied to 55 plasma samples obtained from 34 patients with FD, 5 individuals carrying non-relevant polymorphisms of the GLA gene, and 16 healthy controls. Plasma lyso-Gb3 concentrations were larger in both male and female FD groups compared to healthy subjects (*p* < 0.001). Normal levels of plasma lyso-Gb3 were observed for patients carrying non-relevant mutations of the GLA gene compared to the control group (*p* = 0.141). Dropping the lower limit of quantification (LLOQ) to 0.25 ng/mL allowed us to set the optimal plasma lyso-Gb3 cut-off value between FD patients and healthy controls at 0.6 ng/mL, with a sensitivity of 97.1%, specificity of 100%, and accuracy of 0.998 expressed by the area under the ROC curve (C.I. 0.992 to 1.000, *p*-value < 0.001). Based on the results obtained, this method can be a reliable tool for early phenotypic assignment, assessing diagnoses in patients with borderline GalA activity, and confirming non-relevant mutations of the GLA gene.

## 1. Introduction

Fabry disease (FD) is an X-linked lysosomal disease caused by mutations in the α-galactosidase A gene (GLA), encoding for the homodimeric glycoprotein α-galactosidase A (α-Gal A). In FD, the activity of α-Gal A is heavily reduced, resulting in systemic accumulation of glycolipids, mainly globotriaosylceramide (Gb3) and globotriaosylsphingosine (Lyso-Gb3) [1]. FD shows two major clinical phenotypes, the early onset “classic” form and the later-onset form [2,3]. The “classic” phenotype is characterized by more severe symptoms in male patients, due to the reduced or absent activity of α-Gal A [4]. Affected males exhibit early manifestations such as acroparesthesias, angiokeratoma, corneal opacities, and hypohidrosis starting in childhood or adolescence, which develop into cardiomyopathy, kidney failure, and premature strokes in adulthood, due to the progressive accumulation of glycosphingolipids. The late-onset phenotype shows less severe manifestations and significant residual α-Gal A activity, typically lacking early symptoms but presenting with cardiomyopathy and chronic kidney disease in adulthood [5]. Female Fabry phenotypes are more heterogeneous than male Fabry phenotypes, even among patients carrying the same genotype; they range from asymptomatic to severe manifestations of the disease, depending on random X-chromosomal inactivation during early embryonic development [6,7,8].

Plasma Gb3 has been used extensively as a biomarker of FD [9,10,11]. However, plasma Gb3 quantification has limitations, as most later-onset Fabry males and Fabry females may not show any Gb3 accumulation in plasma; therefore, it provides poor sensitivity as a biomarker [12,13]. On the other hand, the deacylated form of Gb3, lyso-Gb3, has been reported to be the most promising secondary screening target for the diagnosis of Fabry patients, especially later-onset and female patients [13,14,15,16,17,18,19], and a useful marker to evaluate therapeutic efficacy of enzyme replacement therapy (ERT) [20,21,22,23,24,25,26].

Several quantification methods for plasma lyso-Gb3 have been published throughout the past decade, often consisting of time-consuming liquid–liquid extractions (LLE) [14,15,18,27,28], solid-phase extraction (SPE) steps [24,29,30], poorly sensitive protein precipitation-based methods (PPT) [31,32], or extraction protocols involving harmful solvents [33,34]. Here, we present a simple and sensitive method for the analysis of plasma lyso-Gb3 by liquid chromatography–tandem mass spectrometry (LC-MS/MS), with a rapid sample preparation consisting of assisted protein precipitation with Phree cartridges, and successfully validation using lyso-Gb3-D7 as an internal standard. The method was applied to a set of plasma samples from FD patients and healthy controls, along with subjects carrying a non-pathogenic polymorphism [35]. Based on the results obtained, the reported method can be used as a reliable tool for FD diagnosis and to clarify diagnoses in patients with borderline enzymatic α-Gal A activity, or to confirm non-pathogenic polymorphisms of the GLA gene.

## 2. Materials and Methods

### 2.1. Chemicals and Reagents

Analytical standard lyso-Gb3 and its labeled analogue lyso-Gb3-D7 were provided by Merck (Darmstadt, Germany). LC-MS grade methanol was also provided by Merck; UPLC-MS grade acetonitrile and water were provided by VWR chemicals (Radnor, PA, USA). LC-MS grade formic acid (FA) was provided by Carlo Erba Reagents S.r.l. (Milan, Italy). Phree cartridges were provided by Phenomenex (Torrance, CA, USA). Pooled drug-free lithium heparin human plasma for the preparation of quality control (QC) and calibration samples was obtained from the blood bank of the Maggiore Hospital of Bologna, stored at −20 °C and thawed at room temperature before use.

### 2.2. Instrument and Conditions

Quantitative analysis was performed using a triple quadrupole turbo ion spray mass spectrometer (Sciex 4500 QTRAP, Concord, ON, Canada) coupled with a UHPLC system (Nexera X2 UHPLC, Shimadzu Corporation, Kyoto, Japan) equipped with a Kinetex C18, 30 × 4.6 mm I.D., 2.6 μm 100 A column (Phenomenex), protected by a SecurityGuard™ ULTRA Cartridges UHPLC C18 4.6 mm ID (Phenomenex). The analytical column was maintained at a temperature of 30 °C during analysis. Chromatographic separation was achieved over a 3 min linear, binary gradient with a constant flow rate of 0.6 mL/min as follows: 0–1 min, 30% eluent B; 1–4 min, 30–95% eluent B; 4–5 min, 95% eluent B; 5–5.5 min, 95–30% eluent B; 5.5–6.5 min, 30% eluent B. The mobile phases consisted of (A) 0.1% FA in water and (B) 0.1% FA in acetonitrile. The MS/MS analyses were carried out using multiple reaction monitoring (MRM) positive ionization mode. The ion spray voltage was set at 5000 V. The curtain and collision gas (nitrogen) pressures were set at 30 PSI; the nebulizer and heater gas (air) pressures were set at 40 PSI. The ion spray probe temperature was set at 450 °C. The declustering, collision, and entrance potentials were 80, 40, and 13 V, respectively. The monitored transition pairs were 786.46 *m*/*z* > 282.27 *m*/*z* for lyso-Gb3 and 793.56 *m*/*z* > 289.34 *m*/*z* for the IS.

### 2.3. Stock Solutions and Standards

Lyso-Gb3 and the IS were dissolved in pure LC-MS grade methanol, at concentrations of 500 μg/mL and 100 μg/mL, respectively. Appropriate dilutions of the lyso-Gb3 stock solution were made with methanol to prepare calibration curves and QC samples (0.25, 0.5, 50, 100 ng/mL). The IS stock solution was diluted with 0.1% FA in methanol to a concentration of 5 ng/mL and used as a deproteinizing solution. All solutions were stored at −20 °C. Pooled blank human plasma was spiked with lyso-Gb3 working solutions to obtain an eight-point calibration curve (0.25, 0.5, 1, 5, 10, 20, 50, 100 ng/mL). For method validation, QC samples at four concentrations were likewise prepared from pooled blank plasma (LLOQ: 0.25 ng/mL; low: 0.5 ng/mL; medium: 50 ng/mL; high: 100 ng/mL).

### 2.4. Sample Preparation

A total of 100 µL of plasma was placed in a 1 mL Phree cartridge followed by addition of 400 µL deproteinizing solution containing IS at a concentration of 5 ng/mL. The cartridge was vortexed for 2 min, then placed in a collection tube and centrifuged at 1000× *g* at 4 °C for 10 min. A total of 10 µL of filtrate was injected into the chromatographic system.

### 2.5. Method Validation

Validation was carried out according to the European Medicines Agency (EMA) guidelines. The developed method was validated in terms of selectivity, linearity, sensitivity, precision, accuracy, recovery, matrix effect, carry-over, and stability [36].

### 2.6. Selectivity

The selectivity of the method towards endogenous plasma matrix components was assessed in 6 different pools of blank plasma. The absence of interferences was accepted as a response of less than 20% of the LLOQ for the analyte, and 5% for the IS.

### 2.7. Linearity and Sensitivity

Calibration curves were constructed according to a background subtraction approach, as lyso-Gb3 is an endogenous plasma compound. The lyso-Gb3 background peak area in the pooled matrix was subtracted from the area of the peak resulting from added lyso-Gb3. The calculated areas were used to construct the calibration curves [37]. The analyte-to-IS peak area ratios were plotted against the analyte-matched concentration added to the blank plasma. The calibration curves were calculated by the least square method using a weighting factor of 1/x. Linearity was assessed by determining the coefficient of correlation R of the points of the curves. The acceptance requisite was r ≥ 0.998. The back-calculated concentrations of the calibration standards had to be within ±15% of the nominal value. Plasma concentrations of lyso-Gb3 were expressed in ng/mL. LLOQ was defined as the lowest quantifiable concentration of analyte, with associated precision and inaccuracy ≤ ±20% and a signal-to-noise ratio of 10:1. The lower limit of detection (LLOD) was determined in triplicate by comparing measured signals from blank plasma samples spiked with known low concentrations of lyso-Gb3 with those of blank plasma, and calculated as 3 times the baseline noise.

### 2.8. Precision, Accuracy, and Recovery

The precision of the method was assessed by determining the relative standard deviation (RSD) at the four plasma lyso-Gb3 concentrations of QCs, within the same analysis (*n* = 6, intra-assay precision) and in triplicate over a series of three analyses (*n* = 9, inter-assay precision). The accuracy of the method was determined by comparing the means of the calculated concentrations in the above-mentioned QCs with the nominal concentrations (percentage differences) both within the same analysis (*n* = 6, intra-assay accuracy) and in triplicate over a series of three analyses (*n* = 9, inter-assay accuracy). The acceptance criteria for intraday and interday precision and accuracy were ≤±15% for low, middle, and high QCs and ≤±20% for LLOQ. The absolute recovery of lyso-Gb3 from the extraction procedure was determined in triplicate at each QC level, as the ratio between the mean peak area of the analyte-spiked samples and the mean peak area in post-spiked samples. The absolute recovery of IS was similarly estimated at a single concentration of 5 ng/mL.

### 2.9. Matrix Effect and Carry-Over

Matrix effect was also calculated according to the background subtraction approach, and was investigated at low and high QCs, using 5 lots of blank plasma samples from individual donors. For each lot of the matrix, the matrix factor (MF) was calculated as the ratio of the peak area in post-spiked samples to the peak area of neat standard solutions. Matrix effect for the IS was estimated at a single concentration of 5 ng/mL. Carry-over evaluation was performed in each analytical run (*n* = 3) by injecting “blank” samples after the highest calibrator. Carry-over was considered acceptable if it was not greater than 20% of the LLOQ for lyso-Gb3 and 5% for the IS.

### 2.10. Stability

To assess short-term stability, QCs (low and high) were left at room temperature for 2 h, then were processed and analyzed. To evaluate the autosampler stability of processed samples, low and high QCs were stored at 4 °C for 24 h in the instrument autosampler and analyzed in triplicate. QC samples were analyzed against a freshly prepared calibration curve. Processed QCs were considered stable if their mean concentration was within ±15% of the nominal concentration. Long-term stability of plasma QCs stored at −20 °C and −80 °C for 30 days was also evaluated. Samples were considered stable if the deviation of the calculated concentrations from the expected concentrations was within ±15%.

### 2.11. Applicability

We evaluated the applicability of the newly developed UHPLC-MS/MS method by processing 55 plasma samples from: patients with FD, patients carrying functional variants of the GLA gene, and healthy subjects. The procedure was approved by the local Ethics Committee (No. 1077-2020-SPER-AUSLBO). Written informed consent was obtained from each subject. Blood samples were collected into heparinized tubes (8 IU heparin/mL blood) and centrifuged at 1700× *g* for 10 min at 4 °C. The plasma aliquot was separated, transferred into test tubes, and stored at −20 °C for lyso-Gb3 analysis.

### 2.12. Statistical Analysis

Descriptive statistics were carried out with Prism GraphPad 8.0.1 (San Diego, CA, USA). A two-tailed non-parametric *t*-test (Mann–Whitney) was used to assess significant differences between groups. A *p*-value < 0.05 was considered significant. Samples with lyso-Gb3 concentration < 0.25 ng/mL were accounted as 0. The optimal cut-off value to differentiate FD patients from healthy subjects and functional variants was determined by receiver operating characteristic (ROC) curve analysis.

## 3. Results

### 3.1. Method Development

Optimal elution of lyso-Gb3 and the IS were obtained under the chromatographic conditions described above, with a retention time of 2.95 min for both analytes. Representative chromatograms of a “blank” plasma and a QC sample are shown in Figure 1a,b, respectively.

### 3.2. Selectivity

No additional peaks due to endogenous substances that could have interfered with the detection of compounds of interest were observed after the injection of the “blank” pools (*n* = 5) (Figure 1a).

### 3.3. Linearity and Sensitivity

Calibration curves for lyso-Gb3 showed a reproducible linear correlation between analyte concentrations and matched analyte-to-IS peak area ratios over the range of 0.25–100 ng/mL. The equation (mean ± SD, *n* = 3) of the regression line was: y = 0.42302 (±0.00534) x + 0.14881 (±0.01650), r = 0.99911 (±0.00085), where x is lyso-Gb3 concentration, expressed in ng/mL; y is the analyte-to-IS peak area ratio, expressed in arbitrary area units; and r is the correlation coefficient. Back-calculated concentrations of all calibration samples were within ±15% of the nominal value. The LLOQ was set at 0.25 ng/mL (Figure 1b). LLOD was set at 0.10 ng/mL.

### 3.4. Precision, Accuracy and Recovery

Data of intraday and interday precision and accuracy are reported in Table 1. Both intra- and inter-assay precision and accuracy were ≤15% for the whole concentration range. The mean absolute recovery of lyso-Gb3 over the calibration range and IS was greater than 90%.

### 3.5. Matrix Effect and Carry-Over

A matrix factor of 15.7% was found for lyso-Gb3 at high QC. No matrix effect was observed for low QC. No carry-over was observed: injection of blank samples immediately after the highest calibrators (100 ng/mL) yielded signals lower than 20% of the LLOQ for lyso-Gb3 and 5% for IS.

### 3.6. Stability

The concentrations of lyso-Gb3 in processed QCs were stable in the autosampler (4 °C) for 24 h. At room temperature, both QCs were stable for 2 h. Finally, lyso-Gb3 proved stable after 30-day storage at −20 °C and −80 °C (Table 2).

### 3.7. Clinical Application

The developed method was successfully applied to the determination of lyso-Gb3 concentrations in 55 plasma samples obtained from 34 patients with FD, 5 individuals carrying non-relevant polymorphisms of the GLA gene, and 16 sex-matched (8 males, 8 females) and age-matched (*p* = 0.334) healthy controls (Table 3).

All involved patients were receiving ERT, except for four subjects in the male FD group and four subjects in the female FD group. One patient from the female FD group was receiving pharmacological chaperone therapy (migalastat). Overall, lyso-Gb3 concentrations in FD samples ranged from 0.50 to 73.13 ng/mL; lyso-Gb3 concentrations in the male FD group were found to be higher than the female group (*p* < 0.001), agreeing with previous published studies [13,14,15,16,17,18,19,27,28,29,30,31,32,33,34]. No overlap was observed between the whole (combined male and female) FD group and the healthy controls group, or the non-pathogenic polymorphisms group (*p* < 0.001). For both individuals with non-pathogenic mutations and healthy controls, the median concentrations were 0 ng/mL (*p* = 0.141). No statistically significant differences were observed in lyso-Gb3 concentrations between Fabry males vs. untreated Fabry males and between Fabry females vs. untreated Fabry females (Figure 2). A chromatogram obtained from one of the patients’ samples is shown in Figure 1c. From ROC curve analysis a lyso-Gb3 concentration value of 0.6 ng/mL proved to be the optimal cut-off between FD patients and healthy subjects, with 97.1% sensitivity and 100% specificity. Accuracy of the test, expressed by the area under the ROC curve, was 0.998 (C.I. 0.992 to 1.000, *p*-value < 0.001). Two of the FD female patients carrying a *p*.Asn215Ser mutation showed a lyso-Gb3 concentration close to the cut-off value. Both patients were receiving therapy: one patient was treated with migalastat, while the other one was receiving ERT.

## 4. Discussion and Conclusions

Assisted protein precipitation using Phree cartridges for the removal of phospholipids with methanol was chosen as the most efficient procedure to prepare samples, among other tested procedures. Removal of the phospholipidic fraction from samples with Phree cartridges reduces ion suppression and matrix complexity at high *m*/*z* values, improving the S/N ratio for the lyso-Gb3 quantifier MRM pair and enabling a further level of analytical sensitivity [38,39]. Selective phospholipid fraction removal was monitored using the 184–184 *m*/*z* transition (ion-source fragmentation) as a qualitative approach. This transition measured the polar head group fragment characteristic of phosphatidylcholines. Combining this sample preparation with the use of commercially available lyso-Gb3-D7 as an isotope-labeled internal standard allowed us to minimize any matrix effect [40]; improved precision and accuracy; and set the LLOQ of the method at 0.25 ng/mL, providing an accurate measure of residual low lyso-Gb3 concentrations in individuals with harmless GLA gene polymorphisms or healthy controls. Compared to other published methods, the presented method combines fast sample processing, by reducing preparation steps, with high sensitivity, whilst avoiding unsafe solvents such as chloroform (Table 4).

Performing ROC analysis of the obtained results, we set an optimal cut-off value, with high sensitivity and specificity, at 0.6 ng/mL. Previously published cut-off values ranged from 0.4 to 2.3 ng/mL [18,31,33,34]. This wide range is often caused by the heterogeneity of the sample size and uneven sex distribution among FD patients and healthy subjects, preventing the performance of effective statistical tests for group comparisons. Furthermore, the choice of the best cut-off value is also affected by the technical specifications of the methods used for the analysis (i.e., LLOQ).

In conclusion, a reliable and simple two-step LC-MS/MS method was developed, validated, and tested to quantify plasma lyso-Gb3. Rapid sample preparation and an acquisition time of less than 6 min per sample would make this method effective for clinical laboratory practice. Sample preparation could be readily implemented with 96-well Phree plates if high throughput was required. Providing great sensitivity and accuracy, this method would be a useful tool in the diagnostic assessment of patients with an atypical FD phenotype or female carriers of classic FD, who might show lyso-Gb3 plasma concentrations slightly above the experimental cut-off value due to residual GalA activity.

## Figures and Tables

**Figure 1 molecules-26-07358-f001:**
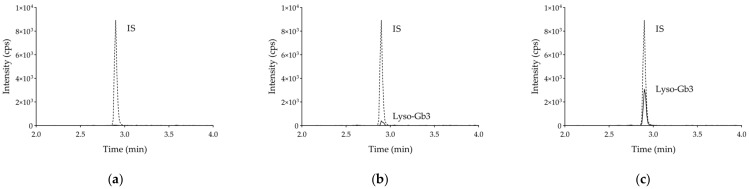
Chromatograms obtained by injecting 10 µL of: (**a**) blank plasma spiked with internal standard; (**b**) blank plasma spiked with lyso-Gb3 at LLOQ (0.25 ng/mL) and internal standard; (**c**) plasma sample of a FD patient spiked with internal standard.

**Figure 2 molecules-26-07358-f002:**
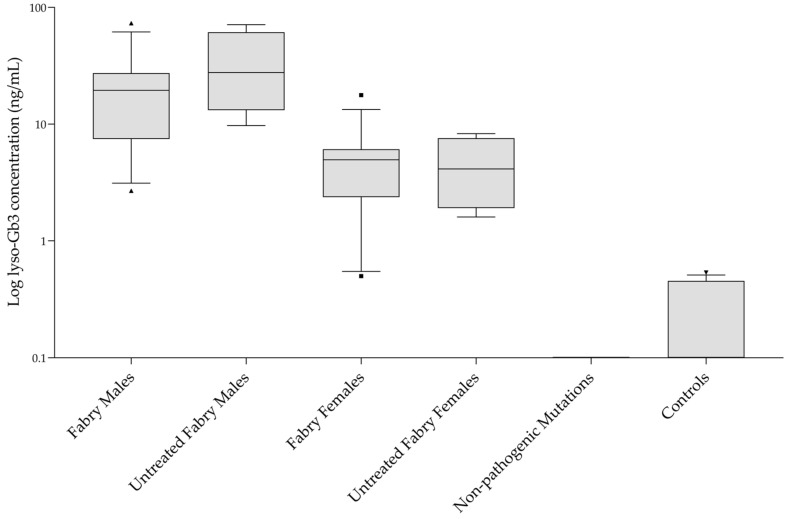
Plasma lyso-Gb3 concentrations for each sample group plotted in log scale. Box plots depict the range between the 25th and 75th percentiles of the data. The horizontal line: median value; capped bars: 10th–90th percentiles; black symbols: outlying values.

**Table 1 molecules-26-07358-t001:** Precision and accuracy of lyso-Gb3 assay.

Lyso-Gb3 Amount Spiked in Blank Plasma (ng/mL)	Intraday (*n* = 6)			Interday (*n* = 9)		
	Calculated Concentration (Mean ± SD) (ng/mL)	Precision (RSD%)	Accuracy (%)	Calculated Concentration (Mean ± SD) (ng/mL)	Precision (RSD%)	Accuracy (%)
0.25 (LLOQ)	0.26 (0.02)	7.7	2.6	0.25 (0.04)	5.6	2.1
0.50	0.53 (0.02)	3.8	5.2	0.52 (0.02)	3.9	3.1
50.00	48.46 (1.30)	2.7	−3.1	49.44 (2.40)	4.8	−1.1
100.00	95.98 (2.48)	2.6	−4.0	94.85 (5.46)	5.8	−5.1

Precision (RSD%): 100 × SD/mean; accuracy (%): 100 × (mean concentration found—known concentration)/known concentration); interday (*n* = 9): triplicate samples for each QC level, over a series of three analyses on different days; LLOQ: lower limit of quantification.

**Table 2 molecules-26-07358-t002:** Stability data of lyso-Gb3 assay.

Lyso-Gb3 Amount Spiked in Blank Plasma (ng/mL)	Stability Test	Mean ± SD (ng/mL) *n* = 3	Precision (%)	Accuracy (%)
0.50	3 h (short stability)	0.52 ± 0.02	3.0	4.6
	24 h (autosampler stability)	0.52 ± 0.03	4.7	5.6
	30 days −20 °C (long stability)	0.53 ± 0.02	3.2	7.1
	30 days −80 °C (long stability)	0.52 ± 0.04	7.9	3.3
100.00	2 h (short stability)	93.68 ± 2.02	2.2	−6.3
	24 h (autosampler stability)	95.93 ± 2.58	2.7	−4.1
	30 days −20 °C (long stability)	106.83 ± 4.05	3.8	6.8
	30 days −80 °C (long stability)	103.29 ± 8.21	7.9	3.3

Precision (RSD%): 100 × SD/mean; accuracy (%): 100 × (mean concentration found—known concentration)/known concentration); LLOQ: lower limit of quantification.

**Table 3 molecules-26-07358-t003:** Demographic and clinical characteristics of subjects.

	Overall FD (*n* = 34)	Male FD (*n* = 17)	Female FD (*n* = 17)	Functional Variants (*n* = 5)	Healthy Subjects (*n* = 16)
Age (years)	46 ± 19	44 ± 20	49 ± 19	38 ± 15	40 ± 15
Lyso-Gb3 (ng/mL)	6.91	20.40	4.95	0.00	0.00
	(3.56–21.22)	(8.79–29.74)	(2.37–6.10)	(0.00–0.00)	(0.00–0.45)

Age presented as mean ± SD. Lyso-Gb3 concentration expressed as median values with 25th and 75th percentile.

**Table 4 molecules-26-07358-t004:** List of published methods for lyso-Gb3 quantification.

Methods	Sample Preparation	No. of Sample Preparation Steps	Calibration Range (ng/mL)	LLOQ (ng/mL)	Recovery (%)
Aerts et al. (2008) [14]	LLE + derivatization	11	0–800	8	>90%
Gold et al. (2013) [27]	LLE	11	2–160	0.04 (LOD)	>98%
Nowak et al. (2018) [18]	LLE	11	0–120	0.3	-
Beasley et al. (2020) [28]	LLE	8	0.4–160	0.4	87%
Boutin et al. (2012) [29]	SPE	9	0.8–320	2	64%
Sueoka et al. (2015) [30]	SPE	11	0–200	0.025	50%
Sakuraba et al. (2018) [24]	SPE	11	0.6–200	-	-
Kruger et al. (2012) [30]	PPT	5	0–400	2.3	100%
Yoon (2015) [32]	PPT	3	2–200	2	98%
Polo et al. (2017) [34]	PPT/CHCl_3_	4	0–800	0.05	-
Talbot et al. (2017) [33]	PPT/CHCl_3_	5	3–1600	4 (LOD)	-
Perrone et al. (2021)	Assisted-PPT	2	0.25–100	0.25	95%

LLOQ: lower limit of quantification; LLE: liquid–liquid extraction; SPE: solid phase extraction; PPT: protein precipitation; CHCl_3_: chloroform.

## Data Availability

The data presented in this study are openly available in Zenodo at doi:10.5281/zenodo.5747798.

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
