# Peer review of "A Rapid and Simple UHPLC-MS/MS Method for Quantification of Plasma Globotriaosylsphingosine (lyso-Gb3)"

_molecules, 2021, doi:10.3390/molecules26237358_

Round 1

Reviewer 1 Report

The manuscript by Alessandro Perrone et al describes rapid and simple methods for quantification of plasma globotriaosylsphingosine (lyso-Gb3) using Phree cartridge and UHPLC-MS/MS.  They first developed the extraction method for lyso-Gb3Cer from plasma using Phree cartridge and adjusted the UHPLC conditions for separation and detection of lyso-Gb3.  They next applied the methods for quantification of lyso-Gb3 from FD patients and healthy controls.  Since plasma lyso-Gb3Cer concentration is one of biomarkers for FD, the method established in this study contribute to diagnosis of FD.  However, the authors need to clarify following points before publishing the Molecules.

Minor points:

Extraction of lyso-Gb3 using Phree cartridge seems to work very well.  Why are phospholipids trapped in the cartridge and lyso-Gb3 recovered from the cartridge?  What is the recovery rate after extraction by the cartridge?

Table 4, to better understand the advantage of the established methods over other published methods, the authors should add parameters for their method in the Table.  Reference number for each method also should be added.

LLOQ should be spelt out.

Reviewer 2 Report

The authors present a well-validated LCMSMS-method for quantification of lyso-GB3 in plasma from Fabry-disease patients and healthy controls, females as well as males.

The LCMSMS method per se is well known, whicht means, it is not new. The only part, which is a novelty in this area is the use of Phree-cartridges in sample preparation, which seems to result in higher sensitivity and accuracy of the method.

Comments:

In the results section, in “clinical application”, you combined GB3-values from untreated patients (females as well as males) with treated ones, which, in my opinion, is not serious, even in a method paper. You should distinguish between these groups, as this will be important for therapy monitoring also, although then sample numbers will decrease even more.

Eventually, longitudinal data, if possible, would be of interest for clinicians, to show whether this method is suitable for therapy monitoring.
